Differentiated historical demography and ecological niche forming present distribution and genetic structure in coexisting two salamanders (Amphibia, Urodela, Hynobiidae) in a small island, Japan

Niwa Keita 1 2 tsushimanda48@gmail.com
Tran Dung Van 3 4
http://orcid.org/0000-0002-6274-4959 Nishikawa Kanto 1 3
1 Graduate School of Human and Environmental Studies, Kyoto University , Kyoto , Japan
2 Akita Prefectural Office , Akita , Japan
3 Graduate School of Global Environmental Studies, Kyoto University , Kyoto , Japan
4 Wildlife Department, Faculty of Forest Resources and Environmental Management, Vietnam National University of Forestry , Hanoi , Vietnam
Nazareno Alison
Electronic publication date: 2022 Apr 28
Publication date: 2022
Volume: 10
Electronic Location ID: e13202
Received 2021 Aug 20; Accepted 2022 Mar 9
Copyright: © 2022 Niwa et al.
Copyright year: 2022
Copyright holder: Niwa et al.
License: This is an open access article distributed under the terms of the Creative Commons Attribution License, which permits unrestricted use, distribution, reproduction and adaptation in any medium and for any purpose provided that it is properly attributed. For attribution, the original author(s), title, publication source (PeerJ) and either DOI or URL of the article must be cited.
License URL: https://creativecommons.org/licenses/by/4.0/

Keywords: Biogeography, Cytochrome b, Ecological niche models, Hynobius tsuensis, Population genetics, Quaternary climate, Tsushima Island, Syntopic

Funding: Environment Research and Technology Development Fund JPMEERF20204002 Project for Human Resource Development Scholarship by Japanese Grant Aid (JDS) This research was performed by the Environment Research and Technology Development Fund (JPMEERF20204002) of the Environmental Restoration and Conservation Agency of Japan. The research of Dung Van Tran in Japan is funded by The Project for Human Resource Development Scholarship by Japanese Grant Aid (JDS). There was no additional external funding received for this study. The funders had no role in study design, data collection and analysis, decision to publish, or preparation of the manuscript.

==============================
Background

The climatic oscillations in the Quaternary period considerably shaped the distribution and population genetic structure of organisms. Studies on the historical dynamics of distribution and demography not only reflect the current geographic distribution but also allow us to understand the adaption and genetic differentiation of species. However, the process and factors affecting the present distribution and genetic structure of many taxa are still poorly understood, especially for endemic organisms to small islands.

Methods

Here, we integrated population genetic and ecological niche modelling approaches to investigate the historical distribution and demographic dynamics of two co-existing salamanders on Tsushima Island, Japan: the true H. tsuensis (Group A), and Hynobius sp. (Group B). We also examined the hypothesis on the equivalency and similarity of niches of these groups by identity and background tests for ecological niche space.

Results

Our result showed that Group A is considered to have undergone a recent population expansion after the Last Glacial Maximum while it is unlikely to have occurred in Group B. The highest suitability was predicted for Group A in southern Tsushima Island, whereas the northern part of Tsushima Island was the potential distribution of Group B. The results also suggested a restricted range of both salamanders during the Last Interglacial and Last Glacial Maximum, and recent expansion in Mid-Holocene. The genetic landscape-shape interpolation analysis and historical suitable area of ecological niche modelling were consistent, and suggested refugia used during glacial ages in southern part for Group A, and in northern part of Tsushima Island for Group B. Additionally, we found evidence of nonequivalence for the ecological niche of the two groups of the salamanders, although our test could not show either niche divergence or conservatism based on the background tests. The environmental predictors affecting the potential distribution of each group also showed distinctiveness, leading to differences in selecting suitable areas. Finally, the combination of population genetics and ecological modeling has revealed the differential demographic/historical response between coexisting two salamanders on a small island.

Introduction

Qualifying the interspecific variation in geographic distribution is central to the understanding of the drivers constraining organism patterns in a region (Brown, Stevens & Kaufman, 1996). Currently, several studies have concentrated on exploring variation in geographic ranges by comparing the ecological niches of given species. The differentiation in ecological niches may provide evidence for evolution and species delineation. Particularly, these species might show niche conservatism, that means the species retain ecological characteristic from common ancestors, leading to allopatric range (Wiens & Graham, 2005) or overlap distribution in secondary contacting (Bett, Blair & Sterling, 2012). Otherwise, niche divergence of two species might be promoted by the reproductive isolation to adapt to contrasting environmental space (Schluter, 2009), leading to occur as sympatric distribution (Via, 2001) or allopatric distribution (Kozak & Wiens, 2006). In some cases, the counter-intuitive result might occur when compared the ecological niches among species due to the heterogeneity of environmental conditions that explain the factors limiting the expansion/contraction of species (Zhang et al., 2014; Tran et al., 2021).

The climatic oscillations in the Quaternary period considerably affected the shape of the distributional pattern and population genetic structure of organisms from boreal, temperate to tropical regions (Hewitt, 2000, 2011). Studies on the historical dynamics of distribution and demography not only reflect the current geographic distribution but also allow us to understand the adaptation of species. The knowledge is also useful for predicting future potential evolution, then proposing the appropriate species management strategies. The empirical studies showed that several species in temperate zone generally showed a trend on contracting population size and/or distribution into refugia and then expanding again (Tominaga et al., 2013; Aoki et al., 2019; Matsui et al., 2020). However, species might respond to the great climatic changes (e.g., the Last Glacial Maximum) by different scenarios (Hewitt, 2000). The process and factors affecting the present pattern and genetic structure of many taxa are still poorly understood, especially for endemic organisms to small islands.

Recently, molecular analysis and ecological niche models (ENMs) at the population level have been integrated as a powerful approach to reveal the historical distribution formation and population dynamics that might shape the current distribution (Chang et al., 2012; Pahad, Montgelard & Jansen Van Vuuren, 2019; Ren et al., 2020). ENMs have been widely used for predicting the ecological niche of species at different scale of space and time. By conjunction the current climatic condition and paleoclimate, the projection of ENMs can present the historically potential pattern and refugia of species during Quaternary period, for example, the Last Glacial Maximum or Mid-Holocene (Aoki et al., 2019; Ren et al., 2020). On the other hand, populational genetic data can reveal genetic diversity and structure of species and estimate the demographic dynamics (Tominaga et al., 2013; Jeon et al., 2021). Consequently, the integration of these approaches has supported each other and shed enlightenment on the evolutionary process of species (Pahad, Montgelard & Jansen Van Vuuren, 2019). However, most of these studies studied a large geographic range and a few studies on a small island were conducted (e.g., Igawa et al., 2013).

The Tsushima Island is a small island (696 km2), located between the Japanese Archipelago and Korean Peninsula, from 34°05′ to 34°42′ N latitude, and 129°10′ to 129°29′ E longitude (Fig. 1A). The island includes two major parts, including Kamijima (northern part) and Shimojima (southern part), and mainly dominated by mountainous geography and a few small plains near the sea (Nagaoka, 2001). Historically, the formation of the straits between Kyushu Island (Japan) and Korean Peninsula was estimated to connect at least five times (Kitamura & Kimoto, 2006). According to Emerson (2002), the biography of species in islands can be clearly explained by natural events in the past. It can be seen that Tsushima Island is a good example to show the biographic dynamics of organisms because of its unique geography, location, and historical connections. In fact, herpetofauna on the island is unique because it is categorized as four types by biogeographic affinities: (1) species distributed on Korean Peninsula, Tsushima Island, and the Japanese Archipelago (the frog Dryophytes japonicus (Matsui & Maeda, 2018)); (2) on the Korean Peninsula and Tsushima Island (the frog Rana uenoi (Matsui & Maeda, 2018), the skink Scincella vandenburghi (Chen et al., 2001), the lizard Takydromus amurensis, the snake Lycodon rufozonatus rufozonatus (Herpetological Society of Japan, 2021)); (3) on Tsushima Island and Japanese Archipelago (the snake Elaphe climacophora (Herpetological Society of Japan, 2021)); (4) endemic to Tsushima Island (the salamander Hynobius tsuensis (Sato, 1943), the frog Rana tsushimensis (Matsui & Maeda, 2018), the viper Gloydius tsushimaensis (Herpetological Society of Japan, 2021)).

Figure 1 The position of Tsushima Island, Japan (A) and sampling localities on Tsushima Island (B).

(B) Red triangles and blue inverse triangles show the collection localities of Group A and Group B, respectively. The shaded areas indicate elevations of 200 m or more.

The Tsushima salamander (Hynobius tsuensis), an endemic salamander to Tsushima Island, thus categorized as (4) noted above, was described by Abe (1922). Recently, Niwa, Kuro-o & Nishikawa (2021) showed that there are two distinct groups within H. tsuensis based on the mitochondrial cytochrome b (cyt b) gene and seven nuclear microsatellite loci, including Group A (probably, true H. tsuensis), and Group B (Hynobius sp.). Group B was phylogenetically closer to Hynobius nebulosus sensu stricto which distributed on Kyushu and its adjacent islands (excluding Tsushima Island) than to Group A (Niwa, Kuro-o & Nishikawa, 2021). Furthermore, the body coloration of the two groups is also distinctive. Group A showed a dark brownish dorsum with yellowish spots or a yellowish dorsum with dark spots, and the upper and lower edges of the tail were bright yellow, while Group B presented a brownish dorsum with dark stipples, and tail without bright yellow color (Niwa, Kuro-o & Nishikawa, 2021). Although the two groups were discovered syntopically from same site (within 5 × 5 m) in the several streams in Tsushima Island, hybrids (including introgressive hybridization) between them were not found. Thus, these two groups (Group A and Group B) were considered to be actually different species (Niwa, Kuro-o & Nishikawa, 2021). Niwa, Kuro-o & Nishikawa (2021) was the first report on the co-existing of two lotic salamanders of Hynobius in a small island. However, the geographic distribution, population genetic, and underlying factors affecting the current distribution of these groups are still unclear.

In this study, we investigated the ecological niche and population genetic of two groups of Hynobius on Tsushima Island that could be useful to infer the fluctuation of historical distribution and demographic dynamics of these two co-existing salamanders on the island. The main objectives of this study include (1) examining the population dynamics between two salamanders from Tsushima Island; (2) elucidating the distribution pattern between two salamanders on Tsushima Island; (3) revealing the ecological niche differentiation of two salamanders on Tsushima Island.

Materials and Methods

Population genetics

Sample collection

Forty-eight specimens of Group A from 27 localities and 20 specimens of Group B from 12 localities were sampled for population genetic analyses (Table S1). Since to be endangered salamanders, we could not collect multiple specimens in several localities. Thus, we sampled one to three specimens per locality in each group, because it was necessary to decrease sampling bias among localities. Tissues were preserved in 96–99% ethanol and stored in freezer at −18 °C. Specimen collection protocols and animal operations followed the guideline of animal experiments in Kyoto University (approval nos. 29-A-7 and 30-A-7). All specimens were stored at the graduate school of Human and Environmental studies, Kyoto University. Total DNA was extracted from muscle or liver tissues using DNeasy Blood & Tissue Kit (Qiagen, Hilden, Germany). Then, all specimens (48 of Group A and 20 of Group B) were sequenced for the partial cyt b gene of 413 base pairs (bp), and methods of PCR, purification of PCR product, and sequencing were the same as those described in Aoki, Matsui & Nishikawa (2013). Cytochrome b gene was widely used for genetic studies on salamanders of Hynobius (e.g., Okamiya et al., 2018; Matsui et al., 2019, 2020; Suk et al., 2020).

Prior to analyses, we added the published sequences of 39 specimens (25 of Group A and 14 of Group B) from DDBJ (DNA data bank of Japan) to our data set. In total, 73 individuals of Group A from 37 localities and 34 individuals of Group B from 17 localities were used in the following analyses (Fig. 1B; Table S1). Of these localities, Group A and Group B were discovered syntopically at nine localities (Fig. 1B; Table S1).

Demographic history

Genetic diversity in each group (Group A and Group B) were calculated with respect to haplotype diversity (h; Wenink, Baker & Tilanus, 1993) and nucleotide diversity (π; Nei & Tajima, 1981). To infer demographic history in each group, we conducted mismatch distribution analysis showing the frequency distribution of pairwise differences (Rogers & Harpending, 1992). We tested if the observed data fit to models of either sudden population expansion or spatial population expansion by calculating goodness of fit on the basis of the sum of square deviations (SSD) with 10,000 replicates. To assess potential expansion, neutrality tests in each group were conducted with 10,000 simulations based on two index, Tajima’s D (Tajima, 1989) and Fu’s Fs (Fu, 1997). Significant negative values of the tests tell us to have been occurred the past demographic expansion. These analyses were performed using Arlequin ver. 3.5 (Excoffier & Lischer, 2010). Furthermore, genetic landscape-shape interpolation analysis in each group were conducted using sequence (partial cyt b gene, 413 bp) and locality data with Alleles In Space (Miller, 2005). The locality data, latitude and longitude were converted into XY coordinates for the analysis. Although this analysis was generally used to visualize spatial patterns of genetic diversity (Miller, 2005; Miller et al., 2006), in the several phylogeographic studies, the analysis has been applied to research past refugia (e.g., Nunome et al., 2010; Tominaga et al., 2013). Finally, we inferred the dynamics of effective population size through time in each group using the Bayesian skyline plot method (Drummond et al., 2005) as implemented in BEAST v2.6.2 (Bouckaert et al., 2019). Using Kakusan4 (Tanabe, 2011), Hasegawa-Kishino-Yano (HKY; Hasegawa, Kishino & Yano, 1985) model was selected as site model for both groups. The substitution rate of cyt b gene sequence for strict clock model was set as 0.64% substitutions per million years (MY) per lineage. Although this value is rate of mitochondrial ND2 gene of Salamandrid (Weisrock et al., 2001), it has widely been applied to phylogenetic studies on Japanese salamanders of Hynobius (Tominaga et al., 2006; Aoki, Matsui & Nishikawa, 2013; Matsui et al., 2017, 2019, 2020; Niwa, Kuro-o & Nishikawa, 2021). An MCMC run for 30 million generations was conducted with sampling every 5,000 generations. Convergence of the parameters was ensured using Tracer ver. 1.6 (Rambaut et al., 2014) with effective sample size (ESS) > 200. After determining burn-in size with Tracer ver. 1.6., the initial 10% generations were discarded as burn-in. The Bayesian skyline plot was reconstructed with Tracer ver. 1.6 to visualize historical demographic pattern.

Ecological niche models

Occurrence data

Present records of two groups were collected from our field survey in 2015–2018. To prevent spatial autocorrelation of occurrence data, we randomly selected points within 100 m, with ten replicates from others by using “spThin” package (Aiello-Lammens et al., 2015). Consequently, 37 localities of Group A (Fig. 2A) and 17 of Group B (Fig. 2B) were employed in the final models.

Figure 2 Upper: A lateral view of the result of a genetic landscape-shape interpolation analysis for Group A (A) and Group B (B). Lower map: Sampling localities of Group A (red triangles) and Group B (blue inverse triangles).

Right and left tips of the upper figures correspond to the northernmost and southernmost localities of Group A (red triangles) and Group B (blue inverse triangles) in Tsushima Island (lower maps). High peaks showed high genetic diversity in the upper figures. Red triangles and blue triangles in the lower maps correspond to “Locality” in Table S1. The shaded areas indicate elevations of 200 m or more.

Environmental variables

Here, we constructed ENMs for the two salamanders using environmental variables from various sources. For climate data, we used 19 bioclimatic data at 30-arc-second (approximately 1 km) resolution from the WorldClim database (Fick & Hijmans, 2017). We also collected LAI (Leaf Area Index), EVI (Enhanced Vegetation Index), and NDVI (Normalized Difference Vegetation Index) in January and June at NASA LPDAAC collection from the MODIS database (https://lpdaac.usgs.gov). To present land cover on Tsushima Island, we gathered the high resolution of Land use and Land cover map products with 30 m resolution from ALOS Science Project (https://www.eorc.jaxa.jp/ALOS/en/lulc/lulc_index.htm). The forest height data was downloaded from the website of the Global Land Analysis & Discovery: https://glad.umd.edu/dataset/gedi/ (Potapov et al., 2021). In addition, the Shuttle Radar Topography Mission (SRTM) at 30 × 30 m resolution (downloaded from https://earthexplorer.usgs.gov/) was used as elevation variables. We, then, calculated the slope and aspect of Tsushima Island from the elevation layer in applying ArcMap 10.6 (ESRI). The variable of pH of water (pH_H2O) of Tsushima Island was gathered from the SoilGrids database that available from ISRIC-World Soil Information: https://soilgrids.org/ (Hengl et al., 2017).

Totally, we collected 31 environmental layers for initial analysis. To avoid autocorrelation among variables, we calculated correlation and subsequently reduced the variable pairs with high correlation (|r| > 0.85) by ENMTools version 1.4.4 (Warren, Glor & Turelli, 2010). Finally, 16 variables were selected for running ecological niche models (Table 1).

Table 1 List of environmental variables.

No.	Name	Sources	Description	
1	Bio02	WorldClim	Mean diurnal range (Mean of monthly = max temp − min temp)	
2	Bio03	Isothermality (BIO2/BIO7) (* 100)	
3	Bio05	Max temperature of warmest month	
4	Bio06	Min temperature of coldest month	
5	Bio12	Annual precipitation	
6	Bio15	Precipitation seasonality (Coefficient of variation)	
7	Elevation	USGS EROS	Height above sea level	
8	Slope	Degree of rise/run	
9	Aspect	Direction a slope face	
10	pH_H2O	Soilgrids	Mean of Soil pH in H2O (at depth: 0–5 cm)	
11	Forest height	GLAD	Forest canopy height (m)	
12	NDVI_Jan	NASA LPDAAC collection	Normalized Difference Vegetation Index of the area in January 2019.	
13	NDVI_Jun	Normalized Difference Vegetation Index of the area in June 2019.	
14	EVI_Jan	Enhanced Vegetation Index of the area in January 2019.	
15	LAI_Jan	Leaf Area Index (LAI) is the one-sided green leaf area per unit ground area in January 2019.	
16	Land Cover	ALOS Science Project	The high resolution of Land use and Land cover map products.	

To project the historical distribution of the salamanders on Tsushima Island, we used climate reconstruction with a Global Climate Models of MIROC-ESM (Sueyoshi et al., 2013) for the Mid-Holocene (~6,000 years ago), the Last Glacial Maximum (LGM) (~22,000 years ago), and the Last Interglacial (LIG) (~120,000–140,000 years BP; Otto-Bliesner et al., 2006), available from the WorldClim database. We kept the data of topography and landcover as constant variables for historical projections because the historical data is only available for climate data at this time.

Ecological niche model processing

We predicted ecological niches for salamanders on Tsushima Island by using Maxent (version 3.4.1; Phillips, Anderson & Schapire, 2006). The method uses present data and environmental conditions to estimate the unknown probability distribution defining a species’ range (Phillips, Dudik & Schapire, 2004; Phillips, Anderson & Schapire, 2006). Comparing to other algorithms, Maxent is shown to perform well, especially for small occurrence sample size (Elith et al., 2006; Wisz et al., 2008; van Proosdij et al., 2016). For Group A, we ran models by using 10-folds cross-validation to evaluate model, while the number of occurrence localities of Group B was limited (n = 17), thus we applied the jackknife method for a small sample size (Pearson et al., 2007). To select the optimal model for the species, we applied the ENMval package in R (Muscarella et al., 2014). The package built a series of model by turning six featured class (L, LQ, H, LQH, LQHP and LQHPT) (L = linear, Q = quadratic, H = hinge, P = product and T = threshold) combined with eight regularizations from 0.5 to 4 (interval = 0.5). In addition, we also adjusted the regularization parameter for each species by ranging from 1 to 10, interval = 0.5 (Bett, Blair & Sterling, 2012), and kept feature class as auto selection. A total of 58 candidate models were analyzed, then, we chose the best model with the minimum value of AICc (Muscarella et al., 2014), and based on the highest AUC (area under the curve; Phillips, Anderson & Schapire, 2006). From candidate models, we selected the best model with regularization value 1 for Group A, and 2 for Group B, and auto feature class. For other parameters, we used default set up as a maximum of 500 iterations, convergence threshold 10−5 (Phillips, Anderson & Schapire, 2006). To determine suitable or unsuitable area, we applied the minimum training presence threshold (the lowest presence threshold) for both species.

Niche equivalency and similarity comparison

Observed niche overlap values for salamanders on Tsushima were calculated by using ENMtools (Warren, Glor & Turelli, 2010) with Schoeners’s D and Hellinger’s I niche similarity metrics. These indices range from 0 (no overlap) to 1 (identical niche models), which predicted the similarity of ecological niche between species (Warren, Glor & Turelli, 2008).

The “Identity test” (also called equivalency test) was used to test whether the ENM of Group A is equivalent to Group B (Warren, Glor & Turelli, 2008). The test creates a null distribution by pooling the occurrence points for both species, randomizing the species identities of the localities, and creating two new samples of the same sizes as the original samples without consideration of suitable habitat to either species. Then, we compared observed niche overlap values to the null distribution of 100 pseudo-replicate niche overlap values by one-side test with an alpha level of 0.05. We considered that these ecological niches were not equivalent if the observed value fell within the bottom 5% of the null distribution. The test was implemented by using the package ENMTools 1.0 (Warren et al., 2021) in R version 4.1.0 (R Core Team, 2021).

We also applied the “background test” in ENMtools version R (Warren et al., 2021) to test for niche conservatism or divergence of two species. The test compared an ENM of Group A against a null distribution generated from random points selected within the geographic range of Group B. We used 100 replicates for the test. The opposite direction, comparing an ENM of Group B to and ENMs generated randomly within the ranges of Group A, also was implemented. Then, the test compared the observed niche overlap value between Group A and B to the null distribution by a two-sided test and alpha level of 0.05. We determined that when the observed niche overlap value between two species was above the 95% confidence interval of the null distribution, it might support niche conservatism. By contrast, the observed value was below the 95% confidence interval, supporting niche divergence (Warren, Glor & Turelli, 2008). In the case of the null hypothesis was supported, the niche overlap might be explained by regional similarities in the habitat available to each group. If the background test is only significant in one direction but not for the remaining direction, we could reject the null hypothesis that similarity (or divergence) between group is less than expected based on the availability of habitat (Warren, Glor & Turelli, 2008).

In addition, we also applied a “PCA-env” framework of Broennimann et al. (2012) that uses kernel density to estimate the environmental distribution of species and the distribution of available environment to quantify niche overlap in a two-dimensional environmental space. Then, the data were used to implement hypothesis tests similar to the approach in Warren, Glor & Turelli (2008) (called ecospat-identity test, and ecospat-background test). Here, we used functions enmtools.ecospat.id and enmtools.ecospot.bg in ENMTools version R (Warren et al., 2021) to conduct the tests. The functions automatically ran principal components analysis to reduce the predictors to a two-dimensional space (Warren et al., 2021) because we employed more than two predictors in the analysis.

Results

Demographic history

Partial cyt b gene sequences (413 bp) were determined for 68 specimens (Group A [n = 48], Group B [n = 20]) and deposited in DDBJ (accession numbers: LC638502–LC638569). Twenty-nine haplotypes and nine haplotypes were observed in 73 individuals of Group A and 34 individuals of Group B, respectively (Table 2). Thirty-one and nine polymorphic sites were detected in Group A (n = 73) and Group B (n = 34), respectively. A3, A24, and [A2, A17, A19] haplotypes were the most (34.2%: 25 of 73 individuals), second (8.2%: six individuals), and third (5.5%: four individuals) dominant haplotypes in Group A, respectively. Also, of 34 Group B individuals, B7, [B3, B6], and B1 haplotypes were the most (23.5%: eight individuals), second (20.6%: seven individuals), and third (14.7%: five individuals) abundant haplotypes, respectively. One (A3) and four (B1, B3, B6, and B7) major haplotypes (frequency within each group >10%) were detected in Group A and Group B, respectively. Twenty and four haplotypes were found from one individual in Group A and Group B, respectively. Genetic diversity was higher in Group A (h ± SD: 0.8702 ± 0.0347; π ± SD: 0.00585 ± 0.00356) than in Group B (0.8520 ± 0.0278; 0.00547 ± 0.00343). Mean number of pairwise differences were larger in Group A (2.4140 ± 1.3256) than in Group B (2.2603 ± 1.2739). One and three differences were the most in Group A and Group B, respectively (Fig. 3). Mismatch distribution test of Group A could not rejected significantly the null hypothesis of population expansion (P > 0.05 in both sudden expansion and spatial expansion models), showed a unimodal shape and fitted the curves of both the sudden expansion (Tau = 1.447, SSD = 0.00035184) and spatial expansion models (Tau = 1.328, SSD = 0.00038387) (Fig. 3). These suggests the occurrence of past demographic expansion in Group A. On the other hands, Group B showed a different result; not suggests the occurrence of population expansion in the sudden expansion model (Tau = 2.910, SSD = 0.02556232, P < 0.05), but suggests that in spatial expansion model (Tau = 2.814, SSD = 0.02246269, P > 0.05) (Fig. 3). In the neutrality tests, Group A showed a significant negative value for both Tajima’s D and Fu’s Fs tests (P < 0.01), which suggested to have been occurred a recent population expansion in this group. On the other hands, results of Group B were not significant for both neutrality tests (P > 0.05) (Table 2).

Table 2 Number of individuals (N), number of haplotypes (Nh), haplotype diversity (h), nucleotide diversity (π), Tajima’s D, and Fu’s Fs.

Group	N	Nh	h ± SD	π ± SD	Tajima’s D	Fu’s Fs	
Group A	73	29	0.8702 ± 0.0347	0.00585 ± 0.00356	−1.97036**	−24.75337**	
Group B	34	9	0.8520 ± 0.0278	0.00547 ± 0.00343	0.08118	−1.43238	
Note:

Double asterisks (**) indicate a significant support (P < 0.01).

Figure 3 Mismatch distribution analyses based on the partial cyt b sequences (413 bp).

Dashed and solid lines indicate the sudden expansion and spatial expansion models, respectively.

Genetic landscape-shape interpolation analysis illustrated geographic patterns of genetic diversity in each group (Fig. 2). The relatively high peaks (i.e., high genetic diversity) were detected at the high mountainous areas in southern part of Tsushima Island for Group A (Fig. 2A), while the results of Group B showed the high diversity in northern part of the island (Fig. 2B).

Bayesian skyline plot presented different curves between Group A and Group B in change of effective population size. It had been increased constantly at least ca. 300,000 to 50,000 years ago in Group A (Fig. 4A). There was no drastic change for the effective population size of Group B during ca. 200,000 to 100,000 years ago, but the occurrence of a recent expansion was estimated from ca. 75,000 years ago to the present (Fig. 4B). Group A had a larger effective population size than in Group B.

Figure 4 The Bayesian skyline plot based on the partial cyt b sequences (413 bp).

Population dynamics of Group A (A) and Group B (B).

Ecological niches of two salamanders

The Maxent models presented strong ability to generate potential distribution of salamanders on Tsushima Island particularly, the AUC for training Group A = 0.895 ± 0.006, and testing AUC = 0.775 ± 0.068, and for Group B was = 0.908 ± 0.008, and testing AUC = 0.908 ± 0.231. The potential distribution of Group A resulting from our models covered most of Tsushima Island, but the high suitable area showed a concentration on the southern part of Tsushima Island. On the contrary, the predicted distribution of Group B was fragmented and mostly restricted to the northern region. The overlap area between the two groups was fragmented and mainly occurred in the northern part of the island (Fig. 5).

Figure 5 The potential distribution of Group A (A), Group B (B) and overlap area between Group A and Group B (C) under current environmental conditions from Maxent model.

The top three variables contributing the model of Group A account for a total of 63.5%, including Bio12 (24.5%), NDVI_Jun (22.4%), and Landcover (16.6%). For Group B, the variables related to topography were the most important, and the contribution of Elevation and Slope were equal with 28.8%. The Bio12 had the third largest contributor to the Group B model at 15.3% (Table 3).

Table 3 The contribution of environmental variables to suitable distribution of Group A and Group B by MaxEnt model.

No.	Group A	Group B	
Environmental variables	% Contribution	Environmental variables	% Contribution	
1.	Bio12	24.5	Elevation	28.8	
2.	NDVI_Jun	22.4	Slope	28.8	
3.	Land cover	16.6	Bio12	15.3	
4.	Elevation	10.6	Forest height	5.3	
5.	Bio5	9.1	pH_H2O	5.3	
6.	LAI_Jan	3.2	Land cover	4.8	
7.	Bio3	3.0	NDVI_June	4.6	
8.	pH_H2O	2.7	Bio6	3.6	
9.	Slope	2.3	Aspect	1.5	
10.	Bio6	2.0	LAI_Jan	0.8	
11.	Aspect	1.2	Bio2	0.3	
12.	Forest height	1.2	EVI_Jan	0.3	
13.	Bio15	1.0	NDVI_Jan	0.2	
14.	NDVI_Jan	0.2	Bio5	0.2	
15.	EVI_Jan	0.0	Bio15	0.2	
16.	Bio2	0.0	Bio3	0.0	

The shared high contributing variables of both groups were Bio12 (annual precipitation with 24.5% for Group A and 15.3% for Group B) and Elevation (with 10.6% and 28.8% for Group A and Group B, respectively). Interestingly, the response curve of the shared predictors presented distinctive trends. In particular, Group A preferred a higher annual precipitation area, roughly 2,050 mm, whereas the most suitable annual precipitation of Group B was around 1,900 mm (Fig. 6). The elevation showed peak suitability around 100 m and 80 m for Group A, and Group B, respectively. However, the habitat suitability of Group A decreased gently while that of Group B presented a significant decrease after peaking (Fig. 6).

Figure 6 Left: The response curves of the annual precipitation (upper) and elevation (lower) of Group A (red solid line) and Group B (blue dashed line). Right: Distribution of the variables on Tsushima Island.

Historical potential distribution

The projected distribution of the two salamanders on paleo-climate reconstructions showed the fluctuation following the time period, and was contrasting compared to the current distribution. Particularly, the potential distributions of both species were increased presently compared to the Last Interglacial, and Last Glacial Maximum (Fig. 7). The distribution of Group A was mostly restricted in the southern part, while the range of Group B concentrated around the isthmus in the center of the island on the Last Interglacial. For the Last Glacial Maximum, the distribution of salamanders tended to move toward the north, Group A focused roughly on the isthmus, whereas Group B moved greatly to the northern tip of the island. On the other hand, the projection on Mid-Holocene increased compared to the present (Fig. 7). Furthermore, it can be seen that the predicted ranges on Mid-Holocene overlapped significantly at the current time for both groups.

Figure 7 The potential distribution of Group A (upper) and Group B (lower) for Last Interglacial (LIG), Last Glacial Maximum (LGM), Mid-Holocene (Mid-Holo), and present scenarios.

Niche equivalency and similarity tests

Our identity test showed that the two salamanders had nonequivalent ENMs (Fig. 8), with Hellinger’s-based I (P < 0.05), and Schoener’s D (P < 0.05), leading to the rejection of the null hypothesis of equivalency test. The background tests indicated that our null hypothesis could not be rejected due to non-significant for both directions of the comparison (P > 0.05 for Hellinger’s-based I; P > 0.05 for Schoener’s D; Fig. 8). The results of ecospat-identity test and ecospat-background test also showed similar trends. The ecospat-identity test rejected the null hypothesis of equivalent niches (P < 0.05) while both directions of ecospat-background test could not reject the null hypothesis of niche similarity (P > 0.05 for Hellinger’s-based I; P > 0.05 for Schoener’s D; Fig. 9).

Figure 8 The results of the identity test, and background tests of Group A and Group B by ENMTools version R.

Black dashed line indicates the results of niche overlap representing the true calculated niche overlap. Red columns show the result of 100 replicates. The left plots showed the Schoener’s D index, and the right plots indicated the Hellinger’s-based I.

Figure 9 The results of ecospat identity test, ecospat background tests of Group A and Group B by ecopat function in ENMtools version R.

Black dashed line indicates the results of niche overlap representing the true calculated niche overlap. Red columns show the result of 100 replicates. The left plots showed the Schoener’s D index, and the right plots indicated the Hellinger’s-based I.

Discussion

Demographic history of two salamanders on Tsushima Island

Both haplotype and nucleotide diversities of Group A was higher than those of Group B in our genetic analysis (Table 2), although number of samples was different between two groups (73 specimens in Group A and 34 in Group B), which was reflected by differences on number of localities (37 localities in Group A and 17 in Group B). ENM analyses also inferred that the suitable habitats of Group A were larger than those of Group B at the Last Interglacial (LIG), Mid-Holocene, and current scenario (Fig. 7), and the results of mismatch distribution analysis (Fig. 3) and neutrality test suggested that Group A have undergone a recent population expansion. Group A seemed to invade (or isolate) into Tsushima Island earlier than Group B (Niwa, Kuro-o & Nishikawa, 2021) and the range is separated into two areas (i.e., the northern and southern areas of Tsushima Island) due to split by the intermediate lowland (Fig. 5A). Further, the genetic landscape-shape interpolation analysis showed that high genetic diversity was detected in the southern part of Tsushima for Group A (Fig. 2A). The two areas are not severely isolated but could provide gene flow between the northern and southern populations of Group A. Thus, Group A retains higher haplotype and nucleotide diversity than Group B. Such high intraspecific genetic diversity may enable Group A much more adapted to the variable climate conditions and elevations than Group B. On the basis of ENM analyses, Group A had a restricted distribution in the Last Glacial Maximum (LGM), but a large highly suitable distribution in the Mid-Holocene (Fig. 7). Further, the highly suitable distribution area under current environmental conditions was continuous within Tsushima Island except for the isthmus between the northern and southern areas (Fig. 5A). These suggest that recent population expansion of Group A rapidly occurred on Tsushima during LGM to the Mid-Holocene, which might be caused by ecological adaptation and its genetic basis, e.g., large genetic diversity in Group A.

On the contrary, Group B has a relatively small distribution range in the current climatic condition (Fig. 7) and our ENM analyses suggested the shrinking of their habitat in the LIG and LGM (Fig. 7). In addition, it seems that Group B has not been split into multiple populations (Fig. 5B). Because Group B was closely related to H. nebulosus from Kyushu Island (Niwa, Kuro-o & Nishikawa, 2021), the ancestral population of Group B is suggested to invade to Tsushima from Kyushu Island relatively recently. If such invasion occurred by small-sized populations, genetic diversity in Group B is expected to be small by the founder effect. Such small genetic diversity will prohibit Group B to expand its range as to cover all the island area.

Results of the Bayesian skyline plot indicated a distinctive population dynamics between two groups; Group A was thought to have increased constantly at least ca. 300,000 to 50,000 years ago, while Group B was considered to have undergone no clear changes in the effective population size during ca. 200,000 to 100,000 year ago and then a recent population expansion during last 75,000 years (Fig. 4). Furthermore, the effective population size of Group A was larger than that of Group B (Fig. 4). These would suggest Group A was more suitable to environment and/or geography of Tsushima Island than Group B, which may have been caused by the earlier invasion (or isolation) into Tsushima in Group A than in Group B.

In the mismatch distribution, Group B was suggested to experience the occurrence of population expansion not in both expansion models, and the frequency of number of pairwise difference was the highest at the three differences (Fig. 3). Also, results of Group B were not significant in the neutrality tests, which means that demographic expansion would not have occurred. On the other hand, occurrence of recent population expansion in Group B was inferred by the result of the Bayesian skyline plot (Fig. 4). Interestingly, Group B has four major haplotypes (B1, B3, B6, and B7), in contrast Group A has only one major haplotype (A3). Group B is thought to have diverged from H. nebulosus relative recently (Niwa, Kuro-o & Nishikawa, 2021), thus it is possible that Group B still has retained ancestral haplotypes to the present. Such ancestral variation may affect the results of mismatch distribution and neutrality tests.

Interestingly, the effective population sizes of both two groups did not decrease around the LGM (ca. 22,000 years ago; Fig. 4), although suitable distribution areas of them shrank at that time (Fig. 7). These mean that population density of two groups was high at the LGM. Salamanders of Hynobius are widely distributed in East Asia and some species occurs on the subarctic zone (e.g., Hynobius retardatus on Hokkaido Island of Japan; Hynobius leechii in the north-eastern China) (Sparreboom, 2014) and up to high mountains at 2,600 m in elevation (Hynobius nigrescens in Honshu Island of Japan; Matsui et al., 2020). Salamanders of Hynobius are cold-resistant animals judged by these facts. Therefore, two groups on Tsushima would have not decreased in the effective population sizes even in the LGM.

Distributional pattern of two salamanders on Tsushima Island

Our results on ENMs of two salamanders from Tsushima Island suggested that both salamanders have their own geographic ranges and unique ecological niches (Figs. 5–7). The potential distribution of Group A was larger (Fig. 5), and suitable range in environmental variables also was broader than those of Group B (Fig. 6), indicating that the Group A had a wider ecological niche of environmental variables compared to Group B. In other words, it means that the Group A was more tolerant than Group B.

The contribution of environmental predictors to models of each group also showed distinctive (Table 3), and the response curves of the shared high contribution variables presented considerably different trends (Fig. 6), indicating that they had different correlation with the set of available conditions. The distribution of Group A was affected by precipitation, vegetation, and land cover, which suggests that Group A has been adapted to climate and biological conditions in Tsushima Island. On the contrary, the distribution of Group B was affected by elevation and slope, which suggests that Group B has selected a given habitat based on topography (i.e., low altitudinal and flat lands). Group B may have been less adapted to the climate and biological conditions in the island because of the relatively recent invasion to the island.

Group A and Group B were found syntopically at nine localities (Fig. 1B), where their larvae inhabited the same mountain streams. Furthermore, two groups were phylogenetically close each other and genetic distance of them was not so large (uncorrected p-distance = 9.2%), although they were not sister species (Niwa, Kuro-o & Nishikawa, 2021). Therefore, ecological habits (e.g., prey item) may be similar with each other. Some biological factors such as competition also may affect the formation of current distributions of two groups. However, the knowledge about competition between two groups is lacked, further studies will be required to clarify factors affecting their distribution patterns.

Our models on historical distribution for the two salamanders suggested a significant contraction during LIG and LGM compared to the current distribution (Fig. 7). Our finding revealed that the distribution of these salamanders might have been affected by the climate change during the Quaternary. The climate in the LIG and LGM were colder and drier than the Mid-Holocene and current conditions (Tsukada, 1983; Takahara & Kitagawa, 2000). Particularity, the global cooling during LGM caused by reduction of sea level and atmospheric CO2 likely led to the smaller potential distribution for both groups than that of current models (Fig. 7). In which, we also found that the suitable area of Group B tends to move to the northern part of Tsushima Island in the LGM. It can be explained by a latitudinal gradient temperature on Japanese Archipelago at this period (Tsukada, 1983). By contrast, the Mid-Holocene model showed a suitable area larger than the predicted by present model for both groups (Fig. 7), supporting by wetter and cooler to warmer climate (Takahara & Kitagawa, 2000; Lutaenko et al., 2007).

Historical refugia of species enable us to expand knowledge on ecological resilience, migration rates in response to shifting climates, and enhance our understanding of how population may react to future climate change (Wielstra et al., 2010). In our result, the genetic landscape-shape interpolation analysis was relatively consistent with the historical suitable area of ENMs, and which suggested refugia used during glacial ages. In Group A, the high genetic diversity and high suitable distribution at the LIG and LGM were projected in southern part (Figs. 2A, 7), suggesting that the past refugia for Group A have existed on the southern part of Tsushima. On the other hand, the relatively high genetic diversity for Group B could be observed in genetic landscape-shape interpolation analysis in northern part of the island (Fig. 2B). The result of the ENM based on LIG and LGM climate projection also showed that suitable areas of Group B concentrated in the northern part (Fig. 7). The location of historical refugia existed within the present distribution of both species suggested that the current ranges of these salamanders were promoted from their refugia during historical ice ages.

Ecological niche differentiation of two salamanders on Tsushima Island

As expected, we found evidence of nonequivalence for the ecological niche of the two groups of the salamanders on Tsushima Island from both identity test and ecospat-identity test (Figs. 8, 9), presenting a lack of exchangeability ecology between them. Otherwise, the result of the background test and the ecospat-background test showed that the null hypothesis was not rejected, meaning the comparison pairs do not show either niche divergence or conservatism based on the background test. The background test could not reveal the divergence/conservatism of the ecological niches between Group A and Group B (Figs. 8, 9), which tells that the niche difference is not so great between them. In fact, the two groups were found syntopically at nine localities from the central to northern parts of Tsushima (Fig. 1B). Probably, another factor may involve determining the distributional difference in the two salamanders. Furthermore, the environmental variables within the selected background sites for these groups were relatively similar, especially for climate data when the area of Tsushima is only a small island (area ~ 696 km2), probably leading to the accepted null hypothesis in background tests. Moreover, nonbiological factors (e.g., level of resolution, or methods of selection of environmental predictors) might also lead to a weak power in the statistic test (Blair et al., 2013). In the present study, it is worth comparing ecology, breeding habits, and life history among syntopic and allopatric areas of the two salamanders. Niwa, Kuro-o & Nishikawa (2021) reported Group B breed earlier than Group A and such difference in breeding habit enable them to occur syntopically. In a preliminary survey by one of the authors (K Niwa, 2015–2018, personal observation), Group A tends to breed in the fast-flowing stream but Group B does in the slow-flowing one, although sometimes they breed in the same stream. Group A might be more adapted to the stream than Group B, which is supported by the wider niche in Group A than Group B and the longer divergence from lentic-breeding ancestors in Group A than Group B in phylogeny (Niwa, Kuro-o & Nishikawa, 2021). The similar result also was detected in other newt species, such as an endemic newt (Lissotriton boscai) in Iberian Peninsula (Peñalver-Alcázar, Jiménez-Valverde & Aragón, 2021), or Lissotriton italicus and L. vulgaris meridionalis in Italian peninsula (Iannella, Cerasoli & Biondi, 2017), or European plethodontid salamanders (genus Hydromantes) (Ficetola et al., 2018). Niche divergence of two groups could be explained by the heterogeneous habitat in environmental space available to each group (Warren, Glor & Turelli, 2008; Blair et al., 2013).

Generally, the body size of closely related organisms correlates with their distribution, species with larger body size possess the widely distributional range due to their competitive relationship in food and/or optimal habitat (Costa et al., 2008; Penner & Rödel, 2019). However, our results showed a different trend when Group A has significantly wider distribution on Tsushima Island, but the body size of two species is mostly equivalent (K Niwa, 2021, unpublished data). Thus, it can be seen that the ecological niche of the two salamanders on the small island were not clearly correlated with their body size. One possible reason to explain the result is the adaptation capacity of each species due to the different time invaded to Tsushima Island. Group A was isolated on Tsushima Island earlier than that of Group B, ca. 3.5 to 3.2 MYA and ca. 1.5 to 1.4 MYA, respectively (Niwa, Kuro-o & Nishikawa, 2021) and occupied mainly the island, including high mountainous area. However, studies on the interspecies interaction and behavior plasticity in both larva and adult period should be conducted to explore the exact mechanisms of ecological relationship between two groups.

Why do two salamanders have the differential demographic/historical patterns?

The present study showed the differential demographic/historical response between two salamanders, although they were mostly sympatric in the small island. One possible reason to explain these differential response between the two groups could be attributed to the different times of invasion to Tsushima Island. Group A is considered to have invaded the island earlier than Group B (Niwa, Kuro-o & Nishikawa, 2021). The different times of invasion may have promoted the different levels of adaptations to the mountainous/lotic environment of the island (i.e., niche differentiation) and those of demographic variation between Group A and Group B. Group A invaded earlier thus to be better adapted in the island than Group B. However, there are no diverse habitats (e.g., breeding site) available to be completely separated by each group in the small (696 km2) and mountainous (a few plains) (Nagaoka, 2001) island. Therefore, it is possible that the ecological niches were not split between two groups and the current sympatric (syntopic) distribution of two salamanders has been retained.

Conclusion

Our study on integrating population genetics and ecological niche modeling suggested the fluctuation in demography and distribution for two co-existing salamanders on Tsushima Island. Results of genetic analyses indicated the different genetic structure and demographic patterns between two salamanders. The population of both salamanders shrank considerably during LIG and LGM, then expanded in Mid-Holocene. The genetic landscape-shape interpolation analysis and ENM results on past climatic projection were consistent in revealing the different historical refugia of these species that probably promoted the present distribution. Our model also predicted the current distribution of Group A mainly focusing on southern part of Tsushima Island, while northern part of Tsushima Island was suitable habitat of Group B. The different effects of environmental predictors to models indicated that each group selects a different set of available conditions. The background tests could not reveal the divergence/conservatism of the ecological niches between the two groups. Thus, we suggested that other factors may involve determining the distributional difference such as micro-habitat selection and interspecific relationship. The result of the study enables us understand distributional and populational dynamics of salamanders in a limited area like Tsushima Island, and may aid the conservation and sustainable management of these unique salamanders.

Supplemental Information

Supplemental Information 1 Partial cyt b gene sequences (413 bp) deposited in DDBJ (LC638502–LC638569).

Click here for additional data file.

Supplemental Information 2 Sampling localities, number of individuals (N), number of haplotypes (Nh), haplotype No., accession No., and sources.

Asterisks indicate that Group A and Group B were discovered syntopically.

Click here for additional data file.

The authors are grateful to Koshiro Eto, Sena Fujii, Ibuki Fukuyama, Kaede Kimura, Tomonori Kodama, Genki Nakatsu, and Tsushima Wildlife Conservation Center for their support in collecting specimens. We thanks to Kazumi Fukutani for her advice on genetic data analysis.

Additional Information and Declarations

Competing Interests

Author Contributions

Animal Ethics

Field Study Permissions

DNA Deposition

Data Availability

The authors declare that they have no competing interests.

Keita Niwa conceived and designed the experiments, performed the experiments, analyzed the data, prepared figures and/or tables, authored or reviewed drafts of the paper, participated in fieldwork, and approved the final draft.

Dung Van Tran conceived and designed the experiments, performed the experiments, analyzed the data, prepared figures and/or tables, authored or reviewed drafts of the paper, and approved the final draft.

Kanto Nishikawa conceived and designed the experiments, analyzed the data, authored or reviewed drafts of the paper, and approved the final draft.

The following information was supplied relating to ethical approvals (i.e., approving body and any reference numbers):

The guideline of animal experiments in Kyoto University (approval nos. 29-A-7 and 30-A-7).

The following information was supplied relating to field study approvals (i.e., approving body and any reference numbers):

There were no locations or organisms that required survey permission in this study.

The following information was supplied regarding the deposition of DNA sequences:

DNA data bank of Japan (DDBJ) accession numbers: LC638502–LC638569.

The following information was supplied regarding data availability:

The detailed locality data, i.e., “latitude” and “longitude” was available for review but cannot be published in order to protect the endangered salamanders (Hynobius tsuensis, Hynobius sp.) from overcollecting by pet-traders.

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
