# Peer review of "Differentiated historical demography and ecological niche forming present distribution and genetic structure in coexisting two salamanders (Amphibia, Urodela, Hynobiidae) in a small island, Japan"

_PeerJ, doi:10.7717/peerj.13202_

## Round 0.1 · original submission · Major Revisions

Dear authors,

I sent the manuscript out for review to determine whether the referees would identify the merits of the study that would justify publication for PeerJ. Although the reviewers recognized merit, they mention drawbacks and limitations, raising some misgivings about data analyses, mainly regarding genetic analyses, and the way the manuscript has been written up. They provided very constructive comments on how the manuscript can be improved. Furthermore, I included comments that should be considered. I hope that you will find all advice helpful when revising the manuscript.

Comments

(1) In the Introduction, authors need to be more precise with the terms used. For instance, what hypothesis (lines 106-107) is being tested? Regarding objective 1 (lines 108-109), what exactly are the authors examining? In addition, authors should update references (not only in the Introduction), focusing on recent papers published.

(2) Authors need to justify the choice of the cytb. In addition, what are the implications of using only one molecular marker?

(3) Methods section (i.e., data analyses) clearly requires more effort to be attractive. In the present form, the lector has a list of results without information about the aim of these analyses. The lector doesn’t know why the authors choose these analyses; ideas must be better structured. Furthermore, a more robust population demographic analysis (e.g., BSP - https://taming-the-beast.org/tutorials/Skyline-plots/) must be done.

(4) As the sample sizes are different among sampling locations and between groups A and B, authors need to be more careful with the comparative analyses. Are the genetic parameters obtained from different sample sizes, comparable among sampling locations or groups? In addition, authors need to justify the sampling and discuss the limitations regarding the small sample sizes employed (e.g., some localities have only one sample).

(5) Overall, the authors should improve the quality of the Figures.

Reviewer 1 ·

Basic reporting

Overall, the manuscript is technically sound and with a good structure. The English language is confusing at times, with several grammatical errors throughout, so the authors need to thoroughly revise the main text.
The figures need some work also. The captions are not self-explanatory and thus make the figures hard to comprehend. For instance, figures 1-2 are incomprehensible to me. Also I would suggest authors to reference the location of the island in the one or more figures using a world/regional map.
The hypothesis being tested is sound (distribution/demographic fluctuations) but the results and analyses are weak and do not give support to this hypothesis.

Experimental design

The research question and methods are ok in general.
The ENMs and niche overlap tests are mostly well done and reported.
However, the genetic analyses are underwhelming and lack detail.
For instance, it is not clear what the sample size is. Line 117 states '48 specimens' but then '68 specimens' are reported in line 123. Also, the genetic landscape-shape interpolation analyses is not explained and thus these results are mostly obscure (see figure 1-2).
In general, the methods need more explanation so the reader can understand how different methods were used and how these are interpreted.
In addition, there are some interesting details on the study system that are only given in the discussion, such as the relationship of group B with another species (H. nebulosus). This is mentioned in line 328, but it would be great having all this background information beforehand. Reading through the text I actually thought of the two groups as sister lineages, just realizing that they are not at the very end.

Validity of the findings

The results of the ENMs are mostly fine, but the genetic analyses are very weak. As the manuscript stands, it reads almost entirely as an ecological niche paper. The genetic diversity analyses are very shallow and are not really contributing to testing the main hypothesis. As such, the manuscript is weak and the main findings do not support the main conclusions.
There is really no effort to integrate genetic diversity with ENMs and the estimation of the basic statistics is not enough to address the question. Overall, as it stands the paper is basically a paper on ENMs and it feels that the population genetics section was a late add on to the manuscript. The ENMs are well explained and conducted but the genetic diversity methods are only briefly explained.
This is exemplified by the fact that something related to population genetics is only mentioned twice in the discussion, and only tangentially (lines 305-306, 369-371). The rest is a discussion of the results of ENMs.
The title mentions genetic structure, but there is really no analyses addressing this. Honestly, the landscape-shape interpolation, although interesting, does not provide anything to the conclusions/discussion.
I would suggest authors think of other type of analyses that can be more powerful to uncover the history of demographic fluctuations and genetic structure. I can think of STRUCTURE-like analyses, mismatch distributions, ABC analyses or simple regressions of genetic diversity vs. ENM stability. This can prove more interesting to uncover interesting patterns (and possibly timing of events). These are just thoughts, and I am not recommending any of these analyses before authors check if their genetic data is sufficient.

Reviewer 2 ·

Basic reporting

Please see my general comments.

Experimental design

Please see my general comments.

Validity of the findings

Please see my general comments.

Additional comments

This is a very interesting manuscript about the niche and range differentiation of two salamanders on Tsushima Island, an island with an interesting geological and geographical history.

I think that the first two paragraphs of the introduction need some re-writing. Ideally, an introduction should start from broader concepts and then narrow down to the specific questions that the authors try to answer, via step-by-step logic transition. The introduction in this manuscript touched on many concepts that are important, but the logic transition is inadequate, leaving the concepts disconnected from one another. This makes it difficult for readers to follow.

I strongly recommend that the authors improve the quality of writing in this manuscript. English is my second language, and I think my academic English level is somewhat representative of an average reader of this paper, as most readers of scientific papers are not from English-speaking countries. In several places, I failed to follow what the authors tried to express. Below I have given a detailed account of the parts of writing that confused me.

I would like to see more discussion on how biotic factors such as competition may have also contributed to the present distribution pattern of the two groups. You mentioned that both salamanders have been found at some localities, but is there any sign of competition between the two species (e.g., competition for resources such as prey items)?

Specific comments:
Line 17-18
Suggest changing “affected considerably on the shaping distributional pattern and population genetic” to “considerably shaped the distribution and population genetic structure of organisms”
Line 21
Suggest changing “pattern” to “distribution”
Line 23
Suggest changing “the population genetics and ecological niche modelling approach” to “population genetic and ecological niche modelling approaches”
Line 16
Should probably be an “of” followed by “similarity”?
Line 29
Change “to occur” to “to have occurred”
Line 29-31
Suggest changing to “The highest suitability was predicted for Group A in southern Tsushima Island”.
Line 36
Should be “nonequivalence”
Line 45-46
Suggest changing “is central research of ecologists to understand” to “is central to the understanding of”
Line 105
“in the contribution” sounds a bit strange. “in this study” is probably better
Line 106
What hypothesis? “Hypothesis” means something very specific in biology. If you have one, state what it is. If you do not have one, the use of “hypothesis” is confusing.
Line 108
Do you plan to introduce both the objectives and the methods used? Then it is better to make the three “objectives” consistent. Currently, the first “objective” is strictly an objective, while the second “objective” is actually a method and the third contains both an objective and the corresponding method.
Line 126
Rather than a table, a figure showing the localities where samples were collected will be much more intuitive for the readers to grasp this information. In my opinion, Table 1 is better suited for the supplementary material.
Line 128
“Totally” should probably be changed to “In total”.
Line 144-145
“we randomly selected points within 100 meters, with ten replicates from others” seems too vague for me to understand that you actually did.
Line 183
“showed to” should be “is shown to”
Line 213
Need to specify which version of R was used.
Line 215-217
I find this sentence very confusing. The way this sentence is structured makes it sound like the goal of the test is to create a “null distribution”, and this is achieved by comparing the ENM of A to a randomly generated ENM. It is beyond me how you can create a null distribution by comparing two things. Furthermore, in ecology, the goal of a test is very often the opposite of what you said, that is, to compare the observed data against the null. Do you mean that “the test compared an ENM of Group against a null distribution generated from random points selected within the geographic range of Group B”?
Line 226-227
What is the null hypothesis? Is it niche conservatism? Please specify.
Line 228-230
I do not understand this sentence because of the grammar errors… Is it “… that uses kernel density to estimate the environmental distribution of species and the distribution of available environment to quantify niche overlap in a two-dimensional environmental space.”
Line 244-246
What is the biological meaning of the negative value? I think most readers including me will appreciate it if you remind them of this.
Line 256
“almost” should be “most of”
Line 326
“that shrinking” should be “the shrinking”
Line 338
What kind of gradient is this? It is too vague for me to understand. Can you elaborate?
Line 340
Should be “is broader”.
Line 342
“tolerant”, not “tolerance”.
Line 350
This seems to hint at competition avoidance in Group B. How can this be concluded from you ENMs? I am not convinced.
Line 352-353
I do not think you can be so certain about this, although it may be a reasonable guess. This uncertainty needs to be reflected in your writing using words such as “may” or “suggest”.
Line 356
“distribution of”
Line 362-363
What gradient? Temperature, precipitation, or something else? Please be specific.
Line 370
I do not understand the use of “resemble” here. Please be more specific.
Line 374
What do you mean by “a tendency for Group B”? Again, please be specific.
Line 381
“nonequivalence”
Line 386-387
This confuses me. What exactly are the null and alternative hypotheses for the background tests?
Line 396
“to compare” should be “comparing”
Line 401
“must be” is too certain a tone here. I do not think you can be this sure (we almost never can in biology).
Line 416
“correlated”, not “correlation”
Line 433-434
Suggest changing “the distinctively selecting the set of available conditions for each species” to “that each species selects a different set of available conditions”
Line 436
“selection”, not “selections”

Reviewer 3 ·

Basic reporting

I read the paper titled “Differentiated historical demography and ecological niche forming present distribution and genetic structure in coexisting two salamanders (Amphibia, Urodela, Hynobiidae) in a small island, Japan”

It deals with a very interesting topic although mainly herpetological. The paper studied the ecological niche of two sympatric salamander gropus/species (one still to be described taxonomically I f I have well understood) providing new data on their population dynamics using genetic information.
The results underline how the two groups/species although sympatric have different distinct areas of high genetic difference and
I found particularly interesting the part dealing with the paleo-climate reconstructions that showed different fluctuations and different ranges compared to the current distribution as the results on the ecological niche of the two groups.

The paper is an important contribution for understanding coexistence mechanism of similar groups/lineages/species in small territories.

Just a major and few minor issues should be addressed to improve the reading of the manuscript.

The main issue is that it is not very clear is the level of sympatry of the two groups and if there is introgression or not among them. Please in the paragraph dedicated to H. tsuensis, detail/describe these aspects.

Also the use of “species”, “groups” “lineages” for the two salamanders studied could be more consistent throughout the text.

Minor points:

Line 63 check if “foundation” is the correct word; I’m not sure of what it means in this context.

Lines 82 -83 the description of the island jumps here a bit unexpected. It could be great to develop a bit more tha narrative of the previous paragraph to increase the connection with the following one

Lunes 92-93 the same for the salamander; maybe it could be possible to state already in the paragraph before that the island is important because inhabited by a peculiar salamander but the species are not two?). In that way the paragraph on the salamander(s) does not fall unexpected for the reader.

Line 101: please clarify which study

Line 104: please maybe it is better to be consistent and use instead of species groups or lineages as made above.
Line 105 : which contribution?
Line 342 check if it should be word or words
Lines 404 -409 it could be possible to refer also to the paper of Ficetola et al., 2018 Differences between microhabitat and broad-scale patterns of niche evolution in terrestrial salamanders Sci. Rep., 10575
Lines 432-434 please rephrase as it I not very clear the meaning of this sentence there is something awkward in it.

Experimental design

The eperimental design was clear nad described in detail

Validity of the findings

Results seem consitent with the methods used

---

## Round 0.2 · Minor Revisions

The authors have done a good job responding to the comments (though not all) pointed out previously. However, there are some minor concerns that still need to be revised as mentioned by the reviewer. I hope that you will find all advice helpful when revising the manuscript.

Reviewer 1 ·

Basic reporting

The English is unclear at times.

Experimental design

no comment

Validity of the findings

no comment

Additional comments

This is my second time reviewing this paper and I think it has improved greatly.
The authors have done a good job at addressing my previous concerns. The methods are much clearer now and the introduction is more fluid.
I believe that the paper is worth publishing and has scientific merit, but also feel like it needs some additional work. The main take away point I'm left with is interesting, but in my opinion this has to be made more explicit, specially in the discussion.
To me this main point would be: the differential demographic/historical response of two mostly sympatric salamander species in a small oceanic island, and why this might be happening.
If authors agree with me that this is the main take-away message, perhaps highlight it in the discussion. This is somehow resumed in lines 434-435, but it is buried in the middle. Why not start here with the main point up front, and then give a more nuanced explanation using the evidence at hand (ENMs, niche tests, demography), and finally discuss possible reasons why this is happening (authors mention different times of colonization).
These are my suggestions to make the results really stand out.
In sum, I think this paper should be published and not much needs to be changed in terms of methods, but also feel that the main point remains somewhat hidden.

Some suggestion on figures: would it be possible to merge figures 2 and 3, so that the comparison between species is more straightforward?
Also, I feel that figure 6 is a bit redundant with figure 8.
Another suggestion for figure 8 to arrange maps differently: four panels on top for group A (left to right: LIG, LGM, HOLO, PRE) and four panels on bottom for group B (same as above). I believe that can help make the visual comparison easier.

Reviewer 2 ·

Basic reporting

Please see my additional comments.

Experimental design

Please see my additional comments.

Validity of the findings

Please see my additional comments.

Additional comments

I did not think I needed to re-review it, as the changes I suggested in the last round of review were minor. Indeed, the authors made all the changes I would like to see.

---

## Round 0.3 · accepted · Accept

Thank you very much for your revised manuscript. Considering the commendable work done in responding to the comments pointed out, I am pleased to recommend acceptance of this manuscript